# Loss of CXC-Chemokine Receptor 1 Expression in Chorioamnionitis Is Associated with Adverse Perinatal Outcomes

**DOI:** 10.3390/diagnostics12040882

**Published:** 2022-04-01

**Authors:** Yin Ping Wong, Noorhafizah Wagiman, Jonathan Wei De Tan, Barizah Syahirah Hanim, Muhammad Syamil Hilman Rashidan, Kai Mun Fong, Naufal Naqib Norhazli, Yashini Qrisha, Raja Norazah Raja Alam Shah, Muaatamarulain Mustangin, Haliza Zakaria, Siew Xian Chin, Geok Chin Tan

**Affiliations:** 1Department of Pathology, Faculty of Medicine, Universiti Kebangsaan Malaysia, Jalan Yaacob Latif, Bandar Tun Razak, Kuala Lumpur 56000, Malaysia; ypwong@ppukm.ukm.edu.my (Y.P.W.); hafizahwagiman@gmail.com (N.W.); barizahsyahirahhanim@gmail.com (B.S.H.); amar@ppukm.ukm.edu.my (M.M.); haliza.zakaria_76@yahoo.com (H.Z.); 2Department of Pathology, Hospital Sultanah Aminah, Johor Bahru 80100, Malaysia; azah2510@yahoo.com; 3ASASIpintar Programme, Pusat Genius@Pintar Negara, Universiti Kebangsaan Malaysia, Bangi 43600, Malaysia; ap02424@siswa.ukm.edu.my (J.W.D.T.); ap02342@siswa.ukm.edu.my (M.S.H.R.); ap02784@siswa.ukm.edu.my (K.M.F.); ap02794@siswa.ukm.edu.my (N.N.N.); ap02433@siswa.ukm.edu.my (Y.Q.); chinsiewxian@ukm.edu.my (S.X.C.)

**Keywords:** chorioamnionitis, CXCR1, interleukin-8, perinatal, placenta

## Abstract

Background: Chorioamnionitis complicates about 1–5% of deliveries at term and causes about one-third of stillbirths. CXC-chemokine receptor 1 (CXCR1) binds IL-8 with high affinity and regulates neutrophil recruitment. We aimed to determine the immunoexpression of CXCR1 in placentas with chorioamnionitis, and its association with adverse perinatal outcomes. Methods: A total of 101 cases of chorioamnionitis and 32 cases of non-chorioamnionitis were recruited over a period of 2 years. CXCR1 immunohistochemistry was performed, and its immunoexpression in placentas was evaluated. The adverse perinatal outcomes included intrauterine death, poor APGAR score, early neonatal death, and respiratory complications. Results: Seventeen cases (17/101, 16.8%) with chorioamnionitis presented as preterm deliveries. Lung complications were more common in mothers who were >35 years (*p* = 0.003) and with a higher stage in the foetal inflammatory response (*p* = 0.03). Notably, 24 cases (23.8%) of histological chorioamnionitis were not detected clinically. Interestingly, the loss of CXCR1 immunoexpression in the umbilical cord endothelial cells (UCECs) was significantly associated with foetal death (*p* = 0.009). Conclusion: The loss of CXCR1 expression in UCECs was significantly associated with an increased risk of adverse perinatal outcomes and could be used as a biomarker to predict adverse perinatal outcomes in chorioamnionitis. Further study is warranted to study the pathophysiology involved in the failure of CXCR1 expression in these cells.

## 1. Introduction

Chorioamnionitis or intraamniotic infection is an inflammation involving the amniotic fluid, foetal membranes, placenta, and/or decidua [1]. It complicates about 1–5% of deliveries at term [2], and about one-third of stillbirths are due to chorioamnionitis [3]. It can be further categorised into clinical and histologic chorioamnionitis [4,5]. The criteria for diagnosis of clinical chorioamnionitis is based on either Gibbs algorithms or the algorithms developed by the expert panel workshop at the National Institute of Child Health and Human Development (NICHD) in the United States. The Gibbs criteria for clinical chorioamnionitis or intraamniotic infection includes maternal fever plus two or more findings of the following: maternal tachycardia, leucocytosis, foetal tachycardia, uterine tenderness or foul odour of the amniotic fluid; while NICHD criteria consists of the presence of maternal fever with one of the following: foetal tachycardia, maternal leucocytosis or purulent fluid from cervical os [2,6,7].

The histological diagnosis of chorioamnionitis depends on the finding of neutrophilic infiltration into the placental plate, chorion and/or amnion. Redline (2006) described the widely accepted histopathologic staging system for inflammation in chorioamnionitis, which is divided into maternal and foetal inflammatory responses (MIR/FIR) [8]. MIR is subcategorised into 3 stages depending on the duration of disease: acute subchorionitis, acute chorioamnionitis and acute necrotising chorioamnionitis. Similarly, FIR is also divided into 3 stages that include foetal chorionic vasculitis or/and umbilical phlebitis, umbilical arteritis, and acute necrotizing funisitis or/and concentric umbilical perivasculitis [8,9,10]. Acute funisitis is a term used to denote the presence of neutrophils within the umbilical cord, and it represents a component of FIR.

Studies showed the occurrence of chorioamnionitis had a bimodal peak at late second/early third trimester and at term [11]. The majority of chorioamnionitis cases are ascending infections and are polymicrobial. The organisms found in the amniotic fluid of patients with chorioamnionitis include *Streptococcus viridans*, *Escherichia coli*, *Clostridium* spp., *Bacteroides* spp., *Ureaplasma* spp., *Gardnerella vaginalis*, *Mycoplasma hominis*, *Lactobacillus* spp., *Pseudomonas aeruginosa*, *Listeria monocytogenes*, and *Campylobacter* spp. [6,12,13,14]. Viral and candida infections, such as severe acute respiratory syndrome coronavirus 2 (SARS-CoV-2), may also give rise to chorioamnionitis [15,16].

Chorioamnionitis was found to be associated with adverse perinatal outcomes, such as foetal growth restriction, intraventricular haemorrhage, respiratory distress syndrome (RDS), bronchopulmonary dysplasia, periventricular leukomalacia, cerebral palsy and poor neurodevelopment, postnatal infection, and neonatal death [17,18,19,20]. Inflammation that involves the foetal chorionic vessels and umbilical cord is of foetal origin. In the umbilical cord, inflammation begins in the vein (phlebitis) and subsequently affects the arteries (arteritis). A gene expression study of tissue obtained from umbilical arteries and veins showed that interleukin 8 (IL-8) expression was higher in the umbilical vein than in the artery [21]. A study reported that histological FIR is observed in one-third of chorioamnionitis cases, and higher neonatal death or impaired neurodevelopment was seen in babies of placenta with acute necrotising funisitis [22].

CXC-chemokine receptor 1 (CXCR1), also called interleukin 8 receptor A (IL8RA), is one of the two high-affinity receptors of IL-8. It is mainly found in the bone marrow, retina, heart, lung, and placenta. It is most abundantly expressed in neutrophils [23]. Moderate expression of CXCR1 is seen in trophoblasts and decidua, and is found to be up-regulated by IL-6 [24,25]. It has chemotactic properties and activates activity on neutrophils.

Intraamniotic inflammation causes a marked increase in IL-8 [17,26]. IL-8 is a highly specific cytokine for neutrophils that binds to CXCR1 and CXCR2. Shimoya et al. reported a significantly higher level of IL-8 by enzyme immunoassay in placentas with chorioamnionitis as compared to those without chorioamnionitis [27]. Immunohistochemical analysis of the placental tissues showed that trophoblasts and macrophage-like cells are IL-8 producing cells. They suggested that IL-8 produced by placental cells might contribute to the immunocompetent ability against invading bacteria. IL-8 was strongly expressed in the term decidua during chorioamnionitis, suggesting that it could be an important regulator of chorioamnionitis-related neutrophil infiltration [28]. Notably, labour itself is an inflammatory state. A study of the gene expression profile of the chorioamniotic membranes obtained from women in labour without chorioamnionitis showed upregulation of neutrophil-specific chemokines, including chemokine C-X-C motif ligand 1 (CXCL1), CXCL2, and CXCL8, and monocyte-specific chemokines C-C motif ligand 3 (CCL3), CCL4, and CCL20 [29].

The study showed that after intraperitoneal lipopolysaccharide injection in mice treated with IL-8 inhibitor to block CXCR1 and CXCR2 receptors, there was a significant reduction in preterm delivery and stillborn. This treated group of mice had less inflammation in the uterine tissue [17]. CXCR1 was also found to be expressed by various cancers, including breast cancer stem cells and malignant melanoma, and blocking it using an anti-CXCR1 (repertaxin) resulted in reduced survival [30,31]. At present, most studies have focused mainly on IL-8, with limited data on CXCR1 expression in chorioamnionitis. The aim of this study was to determine the immunoexpression of CXCR1 in chorioamnionitis. We also studied its association with maternal and foetal inflammatory responses, and with adverse perinatal outcomes.

## 2. Materials and Methods

### 2.1. Study Design

This was a cross-sectional study with a total of 101 placenta samples of mothers with histological chorioamnionitis retrieved from pathology archived records over a period of 2 years. Thirty-two cases of placenta samples of mothers without evidence of histological chorioamnionitis were also included in this study. Cases with insufficient clinical data, incomplete placental tissue, and foetal anomalies were excluded from the study. Clinical data including gestational age, maternal age, parity, ethnicity, clinical diagnosis, and perinatal outcomes (APGAR score, intrauterine death, early neonatal death, pneumonia, respiratory distress syndrome, and other lung complications) were obtained from the integrated laboratory management system. This study was approved by our institutional research ethics committee (approval code: FF-2019-263).

### 2.2. Histological Examination and CXCR1 Immunohistochemistry

All cases were reviewed (NW and GCT) to determine the stage of maternal and foetal inflammatory responses using the criteria described by Redline (2012). One tissue block containing the umbilical cord and membrane roll was chosen for each case for CXCR1 staining. Tissue blocks were sectioned at 3–4 µm thickness and mounted on an adhesive glass slide, Platinum Pro White (Product No: PRO-01, Matsunami Japan). The slides were subsequently incubated with EnVision^TM^ FLEX Peroxidase-Blocking Reagent (Catalogue No. DM821, Dako, Denmark) for 10 min.

Anti-CXCR1 antibody, a rabbit polyclonal (Abcam, MA, USA, Catalogue No. ab137351), was used at a dilution of 1:250. Colon carcinoma tissue was used as a positive control. Immunohistochemical staining was performed following the manufacturer’s protocol from EnVision^TM^ FLEX Mini Kit, high pH (Catalogue No. K8023, Dako Denmark, Glostrup, Denmark). The primary antibody was diluted to optimal concentration using antibody diluent, Dako REAL^TM^ (Catalog No. S2022, Dako Denmark, Glostrup, Denmark). Slides were incubated for 60 min at room temperature with primary antibody, followed by incubation with EnVision^TM^ FLEX Mouse (Linker) (Code No. K8012/K8022, Dako Denmark, Glostrup, Denmark) for 30 min. Incubation with EnVision^TM^ FLEX HRP (Code No. K8023, Dako Denmark, Glostrup, Denmark) for 30 min. Sections were incubated with 1X DAB-containing Substrate Working Solution for 10 min. The slides were counterstained with Hematoxylin 2 (REF 7231, ThermoScientific, Waltham, MA, USA) for 3-dips after the procedures had been completed, followed by dehydration step with increasing alcohol solutions (80%, 90%, 100% and 100%), and twice with Xylene. Finally, the slides were mounted using CoverSeal^TM^-X xylene-based mounting medium (Catalogue No.: FX2176, Cancer Diagnostics, Durham, NC, USA).

### 2.3. Evaluation of CXCR1 Antibody Staining

The scoring of intensity of the immunohistochemical stain was performed by two independent observers (NW and GCT), blinded from the original histologic diagnosis. Whenever there was a disagreement in the result, the slides were reviewed again together, and a consensus was agreed upon. An Olympus microscope BX-41 was used to determine the intensity of immunohistochemical staining in this study. The intensity of CXCR1 staining was scored as negative (0), weak (1), moderate (2), and strong (3). A score of ≥1 was regarded as positive. The percentage of staining was not evaluated, as they were either present diffusely or did not stain completely. CXCR1 expression was evaluated on four placenta cell types: amnion epithelial cells (AEC), decidual cells (DC), umbilical cord endothelial cells (UCEC), and umbilical cord blood vessel smooth muscle wall (UCSMW).

### 2.4. Statistical Analysis

Statistical analysis was performed using either the Statistical Package for Social Science (SPSS for MAC version 26.0, SPSS) or GraphPad QuickCalcs online (https://www.graphpad.com/quickcalcs/catMenu) (accessed on 22 March 2022). Logistic regression, Chi-square, and Fisher exact tests were used in this study. A *p* value of <0.05 was considered statistically significant.

## 3. Results

### 3.1. Demographic Data

This study consisted of 101 cases of histologically diagnosed chorioamnionitis and 32 cases of non-chorioamnionitis. In the chorioamnionitis group, the majority were of Malay ethnicity (75/101, 75.2%), followed by Chinese (20/101, 19.8%), Indian (2/101, 2%), and other ethnicities (3/101, 3%). The mothers’ ages ranged from 24 to 41 years old (average 30.4); most of them were in the 30–39 age group. Twenty-four cases (23.7%) of histologically confirmed chorioamnionitis were not detected clinically. Seventeen (17/101, 16.8%) of the cases with chorioamnionitis presented as preterm deliveries. Table 1 summarises the demographic and clinicopathological characteristics of subjects with and without chorioamnionitis in this study.

### 3.2. Chorioamnionitis with Maternal and Foetal Inflammatory Responses

The majority of the cases with chorioamnionitis were in stage 2 maternal inflammatory response (69/101, 68.3%), followed by stage 1 (22/101, 21.8%) and stage 3 (10/101, 9.9%). Of the 101 cases, 73 had foetal inflammatory response, with most of them in stage 1 (40/101, 39.6%), followed by stage 2 (26/101, 25.7%) and stage 3 (7/101, 6.9%) (Appendix A). There were 6 cases with chorioamnionitis resulting in intrauterine death or neonatal death, and 3 of these cases (3/6, 50%) had histological features of FIR. Five neonates in this study developed lung complications. Four had histological FIR (4/5, 80%; Figure 1; Table 2; *p* = 0.03). In addition, the infants of mothers with chorioamnionitis who were >35 years old had a significantly higher risk of developing lung complications (*p* = 0.003; Table 2).

### 3.3. CXCR1 Expression in Placentas at Different Stages of Maternal Inflammatory Response in Chorioamnionitis and Non-Chorioamnionitis

CXCR1 was expressed in placentas with and without chorioamnionitis. There was no statistically significant difference in CXCR1 expression in the various stages of maternal inflammatory response. The *p* values of 4 types of placental cells (AEC, DC, UBEC, and UCSMW) between MIR stage 0/1 and stage 2/3 were 0.62, 0.63, 0.48, and 0.14, respectively (Table 3).

### 3.4. Logistic Regression Analysis of Foetal Death and Lung Complications as Dependent Variables against Various Independent Variables

The association of different clinicopathological variables with foetal death (miscarriage, stillbirth, and neonatal death) was assessed using logistic regression analysis. The loss of CXCR1 expression on UCEC was significantly associated with foetal death (*p* = 0.009), with 10.7 times more likely to have foetal death than CXCR1-expressed UCEC cases (Table 4, Figure 2). Lower gestational age was also significantly associated with foetal death (*p* = 0.010). Higher stages of MIR and FIR were not significantly associated with foetal death.

In Table 5, the association of different clinicopathological variables with lung complications (pneumonia, respiratory distress syndrome, meconium aspiration, and transient tachypnoea of neonate) was assessed using logistic regression analysis. There was no statistical difference in all the dependable variables, i.e., maternal age, gestational age, MIR stage, FIR stage, and CXCR1 expression in AEC, DC, UCEC, and UCSMW (Table 5).

## 4. Discussion

CXCR1 is a chemokine receptor that plays a crucial physiological role in promoting neutrophilic migration. It is found to be expressed in natural killer cells, lung fibroblasts, lung epithelial cells, and endothelial cells. There is cumulative evidence indicating its involvement in the development of chronic lung disorders and neoplasms [32,33,34]. Various cytokines, such as IL-1, IL-6, IL-8, and tumour necrosis factor-α (TNFα), have been discovered to play a role in the pathogenesis of chorioamnionitis-induced foetal inflammation [35]. A gene expression profiling study of the umbilical cord of extremely preterm newborns with FIR showed upregulation of many inflammatory pathways and molecules, such as cytokines, toll-like receptors, and calgranulins [36].

Chorioamnionitis is a heterogenous disease comprised of different stages of MIR and FIR that could result in various adverse outcomes to both the mothers and newborns depending on its severity. We conducted a logistic regression analysis and showed that early gestational age and loss of CXCR1 expression in UCEC were associated with an increased risk of foetal death (see Table 4). As CXCR1 is important in the activation and recruitment of neutrophils, this suggests that CXCR1 in UCEC is required for the proper regulation of neutrophilic response. The loss of CXCR1 expression may have resulted in a failure to provide sufficient protection to the growing foetus against infection in pregnancy. Kamity et al. showed that blocking CXCR1 resulted in reduced inflammation in the uterus and reduced premature delivery in a liposaccharide-induced mouse model of chorioamnionitis [17].

A previous study showed that a total of 35% of preterm deliveries that had placental inflammation did not demonstrate clinical evidence of chorioamnionitis [37]. Similarly, we noticed that in 23.8% of the cases with histological chorioamnionitis, it was not clinically suspected, and it could have been missed if histologic examination was not thoroughly performed. Erdemir et al. also reported that the majority of histological chorioamnionitis was not detected clinically in their study involving preterm deliveries [18]. In a study conducted in Bangladesh, 12.7% of pregnancies had histological chorioamnionitis; however, only 2.4% were detected clinically [38].

A study showed that neonates of mothers with histological chorioamnionitis had a lower APGAR score, higher incidence of patent ductal arteriosus, bronchopulmonary dysplasia, sepsis, and neonatal death [18]. We found that infants of mothers with chorioamnionitis and maternal ages of more than 35 years old had a significantly higher risk of developing lung complications. Similarly, Kim et al. found that histological chorioamnionitis was associated with a higher risk of interstitial pneumonia, while Du et al. reported that as high as 18% of chorioamnionitis cases were complicated with pneumonia [12,25].

A meta-analysis revealed that the prevalence of pregnant women who had chorioamnionitis was estimated to range from 0.6% to 19.7% [39]. Lahra et al. (2007) reported that 36.9% of stillbirths were associated with chorioamnionitis, and 13.3% of them had FIR [3]. In a preterm study of infants admitted to a neonatal intensive care unit, 32% of the placentas of these infants had FIR [22]. In this study, FIR was identified in 72% of cases with chorioamnionitis. The high number of FIR in our study could be due to the cases referred to us, as a tertiary care centre, being relatively more severe. In this study, logistic regression analyses found no association between high MIR and FIR stages and foetal death and lung complications.

A study reported that preterm infants of mothers with chorioamnionitis and FIR had a higher rate of developing neurodevelopmental impairment and early neonatal death [22]. Zhang et al. reported that infants born to mothers complicated by premature rupture of membrane and FIR had a higher prevalence of brain injury [40]. Furthermore, the cord blood levels of IL-8, TNF-α, and granulocyte colony-stimulating factor (G-CSF) were significantly higher in chorioamnionitis patients with FIR than those without FIR. The diagnosis of histological FIR requires the assessment of the placental foetal surface where the foetal chorionic blood vessels are located as well as the umbilical cord for phlebitis and arteritis. Therefore, it is important to submit the whole placenta, including the umbilical cord, for histological examination. In some countries, the placenta is not completely submitted for assessment due to certain cultural beliefs. As a result, this could limit the accuracy of the interpretation of FIR.

Inflammation acts like a double-edge sword; too little may impair the foetal defence mechanism, while too much could result in uncontrolled tissue damage. Both could eventually lead to devastating consequences. This study is limited by the small number of intrauterine and neonatal deaths.

## 5. Conclusions

We demonstrated that early gestational age and loss of CXCR1 expression in UCEC were associated with an adverse perinatal outcome. Our findings suggest that CXCR1 could be used as a potential biomarker to predict adverse perinatal outcomes in pregnancy with chorioamnionitis. In addition, infants born to mothers under 35 years of age and with a high FIR stage were significantly associated with a higher risk of developing lung complications. Further study is warranted to study the pathophysiology involved in the failure of CXCR1 expression in umbilical cord endothelial cells.

## Figures and Tables

**Figure 1 diagnostics-12-00882-f001:**
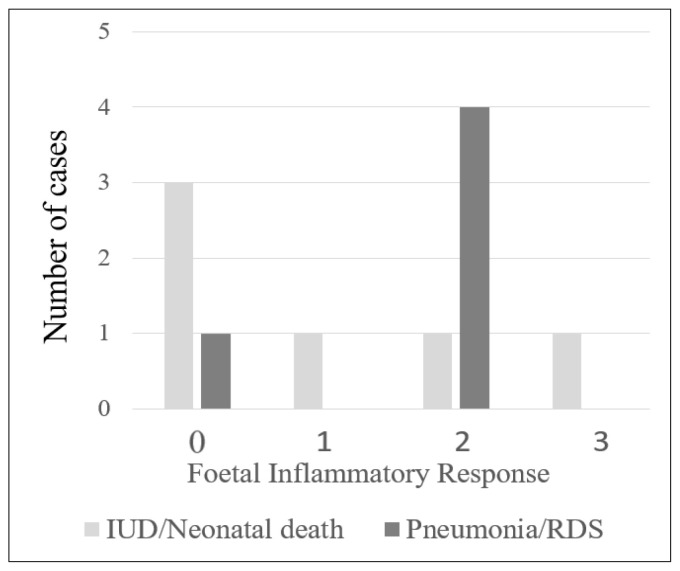
Number of cases with intrauterine death (IUD)/neonatal death (ND) and pneumonia/respiratory distress syndrome (RDS) in relation to histologic foetal inflammatory response.

**Figure 2 diagnostics-12-00882-f002:**
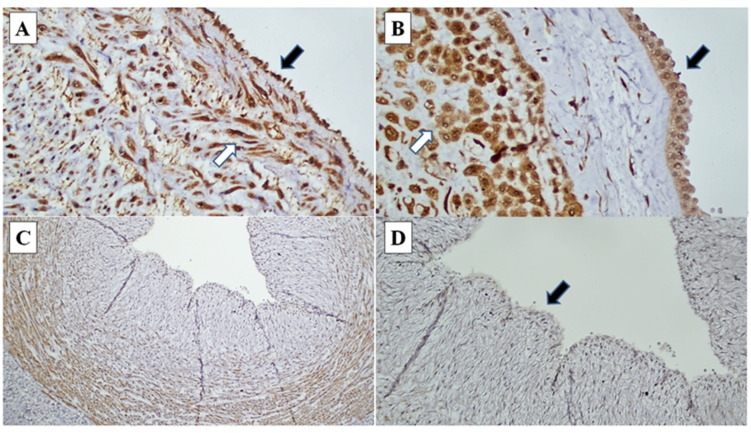
CXCR1 immunohistochemistry in chorioamnionitis: (**A**) Umbilical cord endothelial cells (black arrow) and umbilical cord blood vessel smooth muscle wall (white arrow) demonstrating strong intensity staining (CXCR1, ×200). (**B**) Amnion epithelial cells (black arrow) and decidual cells at decidual capsularis (white arrow), demonstrating strong intensity staining (CXCR1, ×400). (**C**) and (**D**) Loss of CXCR1 immunoexpression in umbilical cord endothelial cells (black arrow) (CXCR1 ×40, ×200).

**Table 1 diagnostics-12-00882-t001:** Demographic data of subjects with and without chorioamnionitis in this study.

	Chorioamnionitis		Non-Chorioamnionitis		*p* Value
Demographic Details	Number of Cases (*n* = 101)		Number of Cases (*n* = 32)		
	No.	%	No	%	
Maternal age (years)					0.057
20–29	46	45.5	8	25.0	
30–39	53	52.5	22	68.8	
40–49	2	2.0	2	6.2	
Ethnicity					0.466
Malay	76	75.2	27	84.4	
Chinese	20	19.8	3	9.4	
Indian	2	2.0	1	3.1	
Others	3	3.0	1	3.1	
Parity					0.004 *
Para 1	59	58.4	8	25.0	
Para 2	23	22.8	10	31.3	
Para 3	14	13.8	9	28.1	
≥Para 4	5	5.0	5	15.6	
Gestational diabetes mellitus					0.342
Yes	17	16.8	7	21.9	
No	84	83.2	25	78.1	
Gestational hypertension					0.425
Yes	1	1.0	1	3.1	
No	100	99.0	31	96.9	
Gestational age					0.31
≤28 weeks	5	4.9	1	3.1	
29–32 weeks	3	3	1	3.1	
33–36 weeks	9	8.9	6	18.8	
≥37 weeks	84	83.2	24	75.0	
Clinical suspected chorioamnionitis					
Yes	77	76.2	NA	NA	
No	24	23.8	NA	NA	

NA—Not applicable.

**Table 2 diagnostics-12-00882-t002:** Correlation between clinicopathological parameters of cases with chorioamnionitis and their perinatal outcomes.

	APGAR Score			IUD/ND			Lung Complications		
	<3	≥3	*p* Value	Yes	No	*p* Value	Yes	No	*p* Value
**Maternal age (years)**			0.35			0.59			**0.003 ***
<35	7	75		6	76		1	81	
35 or more	0	18		0	18		4	14	
**Parity**			1			0.69			1
1	4	55		3	56		3	56	
2 or more	3	39		3	39		2	40	
**Gestational diabetes mellitus**			1			0.59			0.19
Yes	1	16		0	17		2	15	
No	6	78		6	78		3	81	
**Severity of MIR**			0.1			0.18			1
Stage 0/1	0	34		0	34		1	33	
Stage 2/3	7	72		6	73		4	75	
**Severity of FIR**			1.0			1.0			**0.03 ***
Stage 0/1	5	63		4	64		1	67	
Stage 2/3	2	31		2	31		4	29	

FIR—foetal inflammatory response, IUD/ND—intrauterine death/neonatal death, MIR—maternal inflammatory response, * *p* value <0.05 was considered statistically significant.

**Table 3 diagnostics-12-00882-t003:** Comparison of CXCR1 expression at different stages of maternal inflammatory response in cases with chorioamnionitis.

	CXCR1		
Maternal Inflammatory Response	Negative (0)	Positive (1 to 3+)	*p* Value
	AEC		
Stage 0/1	7	42	0.62
Stage 2/3	12	54	
	DC		
Stage 0/1	2	37	0.63
Stage 2/3	2	64	
	UCEC		
Stage 0/1	10	36	0.48
Stage 2/3	13	66	
	UCSMW		
Stage 0/1	3	42	0.14
Stage 2/3	1	78	

Stage 0—No chorioamnionitis, AEC—Amnion epithelial cells, DC—Decidual cells, UCEC—Umbilical cord endothelial cells, UCSMW—Umbilical cord blood vessel smooth muscle wall. Note: Statistical analysis was based on the comparison between stage 0/1 and stage 2/3 of the maternal inflammatory response.

**Table 4 diagnostics-12-00882-t004:** Logistic regression analysis with foetal death as the dependent variable against various independent variables in chorioamnionitis.

Dependent Variable Outcome: Foetal Death				95% C.I. for EXP(B)	
Independent Variables	Regression Coefficient	*p* Value	Odd Ratio	Lower	Upper
Mother′s age	−0.239	0.082	0.787	0.601	1.031
Gestational age	−0.535	**0.010 ***	0.585	0.390	0.878
MIR Stage 1	-	0.332	-	-	-
MIR Stage 2	−19.817	0.998	-	-	-
MIR Stage 3	−1.402	0.137	0.246	0.039	1.565
FIR Stage 1	−0.329	0.791	0.720	0.063	8.197
FIR Stage 2	−1.872	0.206	0.154	0.008	2.802
FIR Stage 3	−1.427	0.337	0.240	0.013	4.412
CXCR1 in AEC	0.916	0.318	2.5	0.414	15.106
CXCR1 in DC	2.944	0.050	19.0	0.996	362.480
CXCR1 in UCEC	−2.367	**0.009 ***	0.094	1.790	63.551
CXCR1 in UCSMW	−18.462	0.999	-	-	-

Foetal death includes miscarriage, stillbirth and neonatal death. MIR—Maternal inflammatory response, FIR—Foetal inflammatory response, AEC—Amnion epithelial cells, DC—Decidual cells, UCEC—Umbilical cord endothelial cells, UCSMW—Umbilical cord smooth muscle wall. * *p* value of <0.05 is considered as statistically significant.

**Table 5 diagnostics-12-00882-t005:** Logistic regression analysis with lung complications as the dependent variable against various independent variables in chorioamnionitis.

Dependent Variable Outcome: Lung Complications				95% C.I. for EXP(B)	
Independent Variables	Regression Coefficient	*p* Value	Odd Ratio	Lower	Upper
Mother′s age	0.132	0.188	1.141	0.937	1.389
Gestational age	−0.028	0.737	0.972	0.824	1.146
MIR Stage 1	-	-	-	-	-
MIR Stage 2	−0.105	0.935	0.900	0.072	11.254
MIR Stage 3	−0.591	0.615	0.554	0.056	5.521
FIR Stage 1	18.638	0.999	-	-	-
FIR Stage 2	-	1.000	-	-	-
FIR Stage 3	19.768	0.999	-	-	-
CXCR1 in AEC	−0.090	0.937	0.914	0.099	8.448
CXCR1 in DC	−18.858	0.999	-	-	-
CXCR1 in UCEC	18.831	0.998	-	-	-
CXCR1 in UCSMW	−18.627	0.999	-	-	-

Lung complications includes pneumonia, respiratory distress syndrome, meconium aspiration and transient tachypnoea of neonate. MIR—Maternal inflammatory response, FIR—Foetal inflammatory response, AEC—Amnion epithelial cells, DC—Decidual cells, UCEC—Umbilical cord endothelial cells, UCSMW—Umbilical cord smooth muscle wall.

## Data Availability

Not applicable.

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
