# Peer review of "Loss of CXC-Chemokine Receptor 1 Expression in Chorioamnionitis Is Associated with Adverse Perinatal Outcomes"

_diagnostics, 2022, doi:10.3390/diagnostics12040882_

Round 1
Reviewer 1 Report
Major concerns fully addressed.
Author Response
Thank you for the comment.

Reviewer 2 Report
The prevalence rate of chorioamnionitis is not well established due
in part to the discrepancies in diagnostic criteria. As symptoms are often subclinical may cause confound of the diagnosis. The signs and symptoms of chorioamnionitis may not become clinically evident until the histologic disease is well established. It is already demonstrated that intra-amniotic inflammation causes a marked increase in interleukin 8, a highly specific cytokine for neutrophils. IL-8 binds to CXCR1 and CXCR2, sustained by a significantly higher level of IL-8 by enzyme immunoassay in placenta with chorioamnionitis as compared to those without chorioamnionitis.
The article aims and succeeds to link CXCR1 and CXCR2 placental expression with chorioamnionitis.
Point 1 I suggest the authors reassess the English language throughout the text.
Point 2 Lines 98 - 100 must be rephrased. In this form, they do not make too much sense.
Author Response
The prevalence rate of chorioamnionitis is not well established due
in part to the discrepancies in diagnostic criteria. As symptoms are often subclinical may cause confound of the diagnosis. The signs and symptoms of chorioamnionitis may not become clinically evident until the histologic disease is well established. It is already demonstrated that intra-amniotic inflammation causes a marked increase in interleukin 8, a highly specific cytokine for neutrophils. IL-8 binds to CXCR1 and CXCR2, sustained by a significantly higher level of IL-8 by enzyme immunoassay in placenta with chorioamnionitis as compared to those without chorioamnionitis.
The article aims and succeeds to link CXCR1 and CXCR2 placental expression with chorioamnionitis.
Response: Thank you for the comments.
Point 1 I suggest the authors reassess the English language throughout the text.
Response: The manuscript has been proofread and all grammatical errors have been rectified.
Point 2 Lines 98 - 100 must be rephrased. In this form, they do not make too much sense.
Response:
Page 3: Study showed after intraperitoneal lipopolysaccharide injection in mice that were treated with IL-8 inhibitor to block CXCR1 and CXCR2 receptors, had a significant reduction in preterm delivery and stillborn. This treated group of mice had less inflammation in the uterine tissue.
Page 10: In some countries, the placenta is not completely submitted for assessment due to certain cultural believe. As a result, this could limit the accuracy of interpretation of FIR.

Reviewer 3 Report
Congratulations for taking the initiative to carry out this interesting research and submitting it to Diagnostics for its possible publication.
This study aims "to determine the immunoexpression of 17 CXCR1 in placentas with chorioamnionitis, and its association with adverse perinatal outcomes.” The subject is interesting but the manuscript suffers from important methodological flaws that must be addressed before proceeding with a more in-depth evaluation.
English need some polish, mainly regarding verbs tenses.
Abstract.
In the “Methods” section the statistical method used is missing.
Admission to the NICU is not a relevant outcome because it is subject to criteria variability.
Chorioamnionitis is associated with an increased risk of premature delivery and with an increased risk of infection, mainly sepsis, in the neonatal period, which is usually greater at lower gestational ages. It is noteworthy that the two outcomes with the greatest clinical relevance have not been taken into account by the authors.
In the section “Results” the sentence “Ninety six percent of cases with chorioamnionitis presented at late third trimester (average 37.2 weeks, median 38 weeks)” is of doubtful meaning because this probably only represents the usual distribution of GA at birth in the center. With some variability, the proportion of full term babies is around 90-93%. In this regard, it is more informative to give the proportion of chorioamnionitis for specific GA. For instance, % of chorioamnionitis in < 28 weeks GA infants, % in 28-32 w GA infants, % in 33-36 w GA, and % in ³ 37 w GA infants.
Introduction.
The sentence (line 58) “Studies showed the occurrence of chorioamnionitis had a bimodal peak at early third trimester and at term” is in dire need of a proper reference.
Material and Methods.
Please correct: Abstract, line 19: “A total of 101 cases of chorioamnionitis and 32 cases of non-chorioamnionitis were recruited over a period of 2 years.”
2.1. Study design, Line 112 “Twelve cases of placenta samples of mothers without evidence of histological 112 chorioamnionitis were also included in this study.”
2.4. Statistical analysis. The method is, in general, quite confusing. Sometimes the FIR is categorized into two (stage 0/1 and stage 2/3). If this categorization is used as the dependent variable, then a logistic regression method is more appropriate than a linear regression method. On the other hand, the meaning of the Coefficient B shown in the tables is not explained. Apparently it is an OR. What is the 95% CI?
In the multivariate linear analysis it is not clear what were de dependent variables and what the independent ones. What confounders were used and what was the rational?
Results.
What was the degree of initial interobserver agreement on the degree of immunohistochemical staining? In how many cases was it necessary an interobserver consensus?
Table 3. If “Statistical analysis was based on the comparison between stage 0/1 and stage 2/3 of maternal inflammatory response”, probably if would be better to show this in the cells, instead of each individual stage (stage 0, 1, 2 and 3) independently, as was shown in Table 2 (stage 0/1 and sage 2/3) to avoid confusion.
In addition, it would be interesting to know how the p-value was calculated. In the case of UCBV (stage 0/1, negative = 3; stage 0/1, positive = 42; stage 2/3, negative =1; stage 2/3, positive =78), the Pearson´s Chi-square is not reliable because there are less than 5 elements in some cells. In this case, the Fisher exact test statistic should be used, which for these numbers, yields a p-value of 0.1354 (not significant), instead of 0,048.
Table 4 is very confusing. “Perinatal outcome” appears as an independent variable. This is not biologically possible, as it is a clear dependent variable.
Author Response
Congratulations for taking the initiative to carry out this interesting research and submitting it to Diagnostics for its possible publication.
This study aims "to determine the immunoexpression of 17 CXCR1 in placentas with chorioamnionitis, and its association with adverse perinatal outcomes.” The subject is interesting but the manuscript suffers from important methodological flaws that must be addressed before proceeding with a more in-depth evaluation.
English need some polish, mainly regarding verbs tenses.
Response: The manuscript has been proofread and all grammatical errors have been rectified.
Abstract.
In the “Methods” section the statistical method used is missing.
Admission to the NICU is not a relevant outcome because it is subject to criteria variability.
Chorioamnionitis is associated with an increased risk of premature delivery and with an increased risk of infection, mainly sepsis, in the neonatal period, which is usually greater at lower gestational ages. It is noteworthy that the two outcomes with the greatest clinical relevance have not been taken into account by the authors.
Response: Thank you for the comments. We agree chorioamnionitis is associated with an increased risk of premature delivery and sepsis. Both has been well described (see references below). Our study does not intend to overemphasise these factors. Furthermore, none of our cases had sepsis. As suggested, we have removed admission to the NICU in the abstract and from table 2. We have also added the following: In this study, 16.8% of cases with chorioamnionitis had preterm deliveries. Also see Table 1 and Page 4.
- Gomez R, Romero R, Edwin SS, David C. Pathogenesis of preterm labor and preterm premature rupture of membranes associated with intraamniotic infection. Infect Dis Clin North Am. 1997;11(1):135-176.
- Shim SS, Romero R, Hong JS, et al. Clinical significance of intra-amniotic inflammation in patients with preterm premature rupture of membranes. Am J Obstet Gynecol. 2004;191(4):1339-1345.
- Gonçalves LF, Chaiworapongsa T, Romero R. Intrauterine infection and prematurity. Ment Retard Dev Disabil Res Rev. 2002;8(1):3-13.
- Hong S, Jeong M, Oh S, et al. Funisitis as a Risk Factor for Adverse Neonatal Outcomes in Twin Neonates with Spontaneous Preterm Birth: A Retrospective Cohort Study. Yonsei Med J. 2021;62(9):822-828.
In the section “Results” the sentence “Ninety six percent of cases with chorioamnionitis presented at late third trimester (average 37.2 weeks, median 38 weeks)” is of doubtful meaning because this probably only represents the usual distribution of GA at birth in the center. With some variability, the proportion of full term babies is around 90-93%. In this regard, it is more informative to give the proportion of chorioamnionitis for specific GA. For instance, % of chorioamnionitis in < 28 weeks GA infants, % in 28-32 w GA infants, % in 33-36 w GA, and % in ³ 37 w GA infants.
Response: Thank you, we agree and have changed the sentence to “Seventeen (17/101, 16.8%) of cases with chorioamnionitis were presented as preterm deliveries.”
Thank you for pointing out the gestational age. We have reanalysed and added the following:
|
Gestational age |
|
|
|
|
0.31 |
|
≤28 weeks 29-32 weeks 33-36 weeks |
5 3 9 |
4.9 3 8.9 |
1 1 6 |
3.1 3.1 18.8 |
|
|
≥37 weeks |
84 |
83.2 |
24 |
75.0 |
|
See table 1, page 4.
Introduction.
The sentence (line 58) “Studies showed the occurrence of chorioamnionitis had a bimodal peak at early third trimester and at term” is in dire need of a proper reference.
Response: Thank you for pointing out. The reference has been added.
Gordon A, Lahra M, Raynes-Greenow C, Jeffery H. Histological chorioamnionitis is increased at extremes of gestation in stillbirth: a population-based study. Infect Dis Obstet Gynecol. 2011;2011:456728.
Figure taken from the article by Gordon et al. 2011.
Material and Methods.
Please correct: Abstract, line 19: “A total of 101 cases of chorioamnionitis and 32 cases of non-chorioamnionitis were recruited over a period of 2 years.”
2.1. Study design, Line 112 “Twelve cases of placenta samples of mothers without evidence of histological 112 chorioamnionitis were also included in this study.”
Response: Thank you for pointing out our mistake. Corrected. Page 3, line 112.
2.4. Statistical analysis. The method is, in general, quite confusing. Sometimes the FIR is categorized into two (stage 0/1 and stage 2/3). If this categorization is used as the dependent variable, then a logistic regression method is more appropriate than a linear regression method. On the other hand, the meaning of the Coefficient B shown in the tables is not explained. Apparently it is an OR. What is the 95% CI?
In the multivariate linear analysis it is not clear what were de dependent variables and what the independent ones. What confounders were used and what was the rational?
Response: Thank you the comments. We have revised the statistical analysis to logistic regression which look at 2 independent variables i.e., 1) Outcome - Fetal death (miscarriage, stillbirth and neonatal death) and 2) Outcome - Lung complications (pneumonia, respiratory distress syndrome, meconium aspiration and transient tachypnea of neonate). The dependable variables were mother’s age, gestational age, MIR Stage 1/2/3, FIR Stage 1/2/3 and CXCR1 in AEC/DC/UCEC/UCSMW. This time, the MIR and FIR stage 1, 2 and 3 were each analysed independently, instead of group into 0/1 and 2/3. The analysis was performed using SPSS.
See Table 4 and 5. Page 8 and 9.
Results.
What was the degree of initial interobserver agreement on the degree of immunohistochemical staining? In how many cases was it necessary an interobserver consensus?
Response: Thanks for the comments. A good point. However, this study does not intend to determine the interobserver variability. Whenever, there was a discrepancy, it was immediately rectified by viewing together and came to a consensus. We did not identify how many disagreements as it was not our intention to evaluate this finding.
Table 3. If “Statistical analysis was based on the comparison between stage 0/1 and stage 2/3 of maternal inflammatory response”, probably if would be better to show this in the cells, instead of each individual stage (stage 0, 1, 2 and 3) independently, as was shown in Table 2 (stage 0/1 and sage 2/3) to avoid confusion.
In addition, it would be interesting to know how the p-value was calculated. In the case of UCBV (stage 0/1, negative = 3; stage 0/1, positive = 42; stage 2/3, negative =1; stage 2/3, positive =78), the Pearson´s Chi-square is not reliable because there are less than 5 elements in some cells. In this case, the Fisher exact test statistic should be used, which for these numbers, yields a p-value of 0.1354 (not significant), instead of 0,048.
Response: Thank you for pointing this out. The stages have been combined as advised. The p value was calculated using Pearson’s Chi-square and when there were <5 elements in the cells, Fisher exact will be used. The calculation was performed using Graphpad online, which does the conversion automatically. Here’s the link the graphpad online: https://www.graphpad.com/quickcalcs/
See Table 3, Page 7.
Table 4 is very confusing. “Perinatal outcome” appears as an independent variable. This is not biologically possible, as it is a clear dependent variable.
Response: Thank you for the comments. Table 4 and 5 were replaced with logistic regression analysis as described above.

Round 2
Reviewer 3 Report
Thank you for the effort in reviewing the work.
There are still some important methodological questions. Please note that in Tables 4 and 5, the dependent variables are mortality and respiratory morbidity, respectively. The independent variables are all the others (maternal age, gestational age, etc.). It is described just the other way around.
Furthermore, it would be interesting to know how "CXCR1 in UCEC" was coded in the logistic regression analysis. Absence is normally coded as "0" and presence as "1". In this case, it is striking that they seem to be coded inversely. As it appears in the table, it suggest that the presence of CXCR1 is associated with higher mortality (values significantly above 1). If its presence is associated with lower mortality, the values and 95% CI should be less than 0. To be sure of what is presented, it would be interesting to know in a 2x2 table the percentages of CXCR1-expresion Yes/No and mortality Yes/No.
Author Response
Thank you for the valuable comments.
Thank you for the effort in reviewing the work.
There are still some important methodological questions. Please note that in Tables 4 and 5, the dependent variables are mortality and respiratory morbidity, respectively. The independent variables are all the others (maternal age, gestational age, etc.). It is described just the other way around.
Response: Thank you for comments. The outcomes (fetal death and lung complications) were changed to dependent variables and the clinicopathological features were changed to independent variables.
See Table 4 and Table 5, Page 8 and 9.
Furthermore, it would be interesting to know how "CXCR1 in UCEC" was coded in the logistic regression analysis. Absence is normally coded as "0" and presence as "1". In this case, it is striking that they seem to be coded inversely. As it appears in the table, it suggest that the presence of CXCR1 is associated with higher mortality (values significantly above 1). If its presence is associated with lower mortality, the values and 95% CI should be less than 0. To be sure of what is presented, it would be interesting to know in a 2x2 table the percentages of CXCR1-expresion Yes/No and mortality Yes/No.
Response:
Thank you for the comments. In the CXCR1 in UCEC in logistic regression analysis, negative is coded as ‘0’ and positive is coded as ‘1’. However, when the reference category for UCEC (indicator) was put as “last”, as a result, we got a positive value of 2.367 (B) and 10.667 (EXP(B)) (see below Table 1). By changing the reference category to “first”, we obtained a negative value of -2.367 (B) and 0.094 (EXP(B)) (Table 2). The odd ratio, in both table 1 and 2 are the same, one divided by 0.094 gives 10.667.
Corrected as suggested. See Table 4.
Table 1
|
Variables in the Equation |
|||||||||
|
|
B |
S.E. |
Wald |
df |
Sig. |
Exp(B) |
95% C.I.for EXP(B) |
||
|
Lower |
Upper |
||||||||
|
Step 1a |
CXCR1 negative and positive(1) |
2.367 |
.911 |
6.758 |
1 |
.009 |
10.667 |
1.790 |
63.551 |
|
Constant |
-3.689 |
.716 |
26.552 |
1 |
.000 |
.025 |
|
|
|
|
a. Variable(s) entered on step 1: CXCR1 negative and positive. |
|||||||||
Table 2
|
Variables in the Equation |
|||||||||
|
|
B |
S.E. |
Wald |
df |
Sig. |
Exp(B) |
95% C.I.for EXP(B) |
||
|
Lower |
Upper |
||||||||
|
Step 1a |
CXCR1 negative and positive(1) |
-2.367 |
.911 |
6.758 |
1 |
.009 |
.094 |
.016 |
.559 |
|
Constant |
-1.322 |
.563 |
5.517 |
1 |
.019 |
.267 |
|
|
|
|
a. Variable(s) entered on step 1: CXCR1 negative and positive. |
|||||||||
Response:
As commented, below is the table generated using SPSS to calculate the 2x2 table based on CXCR1 expression and mortality.
|
Death outcomes * CXCR1 negative and positive Crosstabulation |
||||
|
Count |
||||
|
|
CXCR1 negative and positive |
Total |
||
|
Negative (0) |
Positive (123) |
|||
|
Death outcomes |
No death |
15 |
80 |
95 |
|
Death |
4 |
2 |
6 |
|
|
Total |
19 |
82 |
101 |
|

This manuscript is a resubmission of an earlier submission. The following is a list of the peer review reports and author responses from that submission.
Round 1
Reviewer 1 Report
The main message of the ms. by Wagiman et al. focuses on a strong relationship between the loss of expression of placental CXCR1 and adverse perinatal outcome in histologic chorioamnionitis. Histology and immunochemistry appear to be appropriate, while the description of the clinical perinatal outcomes needs more details. Data analysis is based on non-parametric tests and a quite asymmetrical population comparison (sample sizes n=101 positive vs. n=12 negative for inflammation). As authors should be aware, there are several algorithm to calculate appropriate study Group and control Group for a power of 80% at a p value of <0.05. Furthermore, as described by Fisher in his original work, a p value < 0.05 does means per se nothing that a variable that may merit further investigation. A univariate analysis to identify potential interesting variables followed by a logistic regression analyis is mandatory when dependent over independent clinical variables interact like in this case during pregnancy, intrapartum period and neonatal period. Although the data provided by the AA are interesting, the ms. suffers from a general lack of conciseness and weak architecture, especially in Introduction, Mat & Method and Discussion. I suggest a major style revision for those sections. Language is generally understandable although it needs grammar and style revision. Title and Abstract should be rephrased in order to convey the ms. message.Reviewer 2 Report
The manuscript entitled “Placental CXCR1 Expression in Histological Chorioamnionitis and its Association with Adverse Perinatal Outcomes” is overall interesting and well written.
However, there is a major concern about the original work submitted: how can the Authors explain the CXR1 negativity in stillbirth? Could the negativity be due to loss of expression caused by fetal maceration in utero? How can the Authors differentiate between a true negativity and a false negativity as a result of fetal death?
I would suggest the Authors prepare a different table for intrauterine death and neonatal death specifying the different cells involved. Moreover, in case of stillbirth I would specify the grade of fetal maceration as it affects immunohistochemical expression.
Moreover, 12 cases taken as controls are very few. I would improve the sample of at least 30 cases, as the Authors compare them to 101 cases with chorioamnionitis.
Regarding minor concerns, they are as follows:
- Chorioamnionitis is a singular term; please avoid the verb in plural tense.
- What does AEC, DC, UBEC and 
UCBV 
stand for? Before inserting the abbreviation the Authors should leave the whole word.
- Some minor spelling and grammar mistakes (please, check).